

# Serum proteomics study on cognitive impairment after cardiac valve replacement surgery: a prospective observational study

Huanhuan Ma[1,2,*], Yiyong Wei[3,*], Wei Chen[4], Song Chen[4], Yan Wang[4], Song Cao[5] and Haiying Wang[4,6]

[1] Department of Anesthesiology, Medical College of Soochow University, Soochow, Jiangsu Province, China
[2] Department of Anesthesiology, Zunyi Maternal and Child Health Care Hospital, Zunyi, Guizhou Province, China
[3] Department of Anesthesiology, Longgang District Maternity and Child Healthcare Hospital of Shenzhen City, Shenzhen, Guangdong Province, China
[4] Department of Anesthesiology, The Affiliated Hospital of Zunyi Medical University, Zunyi, Guizhou Province, China
[5] Department of Pain Medicine, The Tenth Affiliated Hospital of Southern Medical University, Dongguan, Guangdong Province, China
[6] Department of Anesthesiology, Zunyi Medical University, Zunyi, Guizhou Province, China
[*] These authors contributed equally to this work.

Corresponding author
Haiying Wang, wanghaiting-8901@163.com

## ABSTRACT

**Objective**. The incidence of perioperative neurocognitive disorders (PND) is high, especially after cardiac surgeries, and the underlying mechanisms remain elusive. Here, we conducted a prospective observational study to observe serum proteomics differences in PND patients after cardiac valve replacement surgery.

**Methods**. Two hundred and twenty-six patients who underwent cardiac valve surgery were included. They were categorized based on scoring into non-PND group (group non-P) and PND group (group P'). The risk factors associated with PND were analyzed. These patients were further divided into group C and group P by propensity score matching (PSM) to investigate the serum proteome related to the PND by serum proteomics.

**Results**. The postoperative 6-week incidence of PND was 16.8%. Risk factors for PND include age, chronic illness, sufentanil dosage, and time of cardiopulmonary bypass (CPB). Proteomics identified 31 down-regulated proteins and six up-regulated proteins. Finally, GSTO1, IDH1, CAT, and PFN1 were found to be associated with PND.

**Conclusion**. The occurrence of PND can impact some oxidative stress proteins. This study provided data for future studies about PND to general anaesthesia and surgeries.

## INTRODUCTION

Perioperative neurocognitive disorder (PND) manifests as postoperative cognitive decline, encompassing memory impairment, abstract thinking difficulties, and orientation disruptions. These symptoms often coincide with reduced social engagement and integration capabilities (*Evered et al., 2018*). Degenerative diseases have become increasingly common among elderly individuals, including valvular heart disease. At the same time, heart valve replacement surgery has been increasing year by year (*Hao, Hei & Hou, 2021*). Age is the main independent risk factor for PND (*Kubota et al., 2018*). Cognitive dysfunction occurs in 7% to 26% of patients after non-cardiac surgery (*Moller et al., 1998*). However, this prevalence rises dramatically from 14% to 60% following cardiac surgeries (*Keizer et al., 2005*). Previous research has demonstrated that proteins Beta-amyloid and Tau in cerebrospinal fluid are linked to the underlying mechanism of cognitive impairment (*Blennow & Zetterberg, 2018*; *Hansson et al., 2019*).

Present research illustrates that PND's pathogenesis is multifaceted, including encompassing oxidative stress (*Netto et al., 2018*), neuroinflammatory responses (*Luo et al., 2019*), mitochondrial dysfunction (*Rivero-Segura et al., 2019*; *Salminen et al., 2012*), blood–brain barrier injury (*Marungruang et al., 2018*) and synaptic damage (*Xiao et al., 2018*). Given this complexity, it is insufficient to rely solely on low concentrations of beta-amyloid proteins in the cerebrospinal fluid and elevated levels of phosphorylated tau proteins for diagnosis and treatment (*VanDusen et al., 2021*). Studies are needed to identify and utilize more proteins to enhance diagnostic precision and therapeutic effectiveness.

This study aimed to use proteomics to explore potential PND-related proteins by analyzing serum samples from patients undergoing post-cardiopulmonary bypass (CPB) valve replacement. This study may offer new PND research targets.

## METHODS

### Participants

We conducted a single-center prospective cohort study that was approved by the Biomedical Research Ethics Committee of the Affiliated Hospital of Zunyi Medical University on October, 2022 (KLL-2022-700) and was registered in the Chinese Clinical Trials Registry on October, 2022 (ChiCTR2200064929). Patients who underwent heart valve replacement at Zunyi Medical University Hospital from October 2022 to April 2023 under CPB were included (the first patient in this study was included on October 11th, 2022, and the last visit was on November 24th, 2022). Written informed consent was obtained prior the enrollment. The inclusion criteria were as follows: Patients between the ages of 45 and 74 who underwent heart valve replacement surgery under CPB and are classified as ASA grade I- III. The exclusion criteria: surgical history, cerebrovascular disease, epilepsy, other central nervous system diseases, metabolic disease, history of endocrine disorders, patients with neurological or psychiatric illnesses, preoperative Mini-Mental State Examination (MMSE) score $\leq 23$, and emergency patients.

## Anesthesia and surgery

No preoperative medication was given. Anesthesia was induced with midazolam (2 mg), sufentanil (0.4~0.6 µg/kg), atracurium cisbenesulfonic acid (0.2~0.3 mg/kg) and etomidate (0.3 mg/kg). Anesthesia maintenance involved the intravenous pumping of sufentanil (0.5~1.0 µg/kg/h), cisatracurium (0.2 mg/kg/h), propofol (0.5~2 mg/kg/h). Ventilator parameters were adjusted according to airway pressure and end-expiratory $CO_2$ ($ETCO_2$) was maintained at 35~45 mmHg. Invasive arterial blood pressure (mmHg), body temperature (T), heart rate (HR), saturation of pulse oxygen ($SPO_2$), electrocardiogram (ECG), bis-frequency index (BIS) and $ETCO_2$ were monitored. Intraoperative mean arterial pressure (MAP) and HR fluctuations were maintained within $\pm20\%$ of baseline values.

Patients were returned to the Intensive Care Unit (ICU) on the vascular surgery ward with a tracheal catheter after surgery. Postoperative analgesia was managed in a uniform manner in the ward. The use of opioids and vasoactive drugs was recorded. Two milliliters peripheral venous blood was collected on the second day post-surgery after neuropsychiatric scale assessment. Blood samples were centrifuged at 4 °C for 3,000 rpm for 15 min. Serum was taken and stored in a refrigerator at −80 °C.

## Neuropsychological test

MMSE (provided by The Hartford Institute for Geriatric Nursing, Division of Nursing, New York University) and MoCA-B evaluation scales were performed on pre-surgery and post-surgery, and scores for each test were recorded by trained investigators. Z-score was calculated for each patient based on the scores from pre-surgery and post-surgery cognitive function tests. Cognitive impairment testing was performed with consideration of potential learning effect. The means change from the health control group (spouse of patients) was subtracted to correct for this learning effect. This means that the preoperative score and learning effect were deducted from each postoperative test score. The difference was then divided by the preoperative standard deviation (SD) of the healthy control group. The composite Z-score was calculated as the average Z-score of all tests in a single patient. PND was determined by a Z-score of $<-1.96$ in two or more test items (*Greaves et al., 2020*; *Wiberg et al., 2022*).

$$Z = \frac{\text{Postoperative Score} - \text{Preoperative Score} - \text{Learning Effect}}{\text{Preoperative Standard Deviation}}.$$

## Grouping

Based on Z-scores, patients were categorized into PND group (group P') and non-PND group (group non-P). Using PSM (with four risk factors as covariates and a caliper width of 0.2) to match 1:1 from the group non-PND to create a matched group C that has similar baseline characteristics to the group P' (*Wang, 2021*). The resulting matched groups are the group P and the group C.

## Sample size

The postoperative patients were divided into two groups: PND (group P') and non-PND (group non-P). The incidence rate of PND was evaluated as approximately 24% based

on preliminary experiment. The reported incidence of neurocognitive disorders after cardiac valve surgery in Chinese individuals was approximately 33% according to relevant literature (*Rappold et al., 2016*; *Xu et al., 2013*). Using the PASS 2021 software (Kaysville, Utah, USA) and setting the parameters for one proportion, two sides to P0 0.33, P1 0.24, with a significance level of 0.05 and a test power of 0.80, we estimated a drop-out rate of 10% and included 226 subjects in the study.

## Proteomics

After matching, serum samples from groups P ($n = 25$) and C ($n = 25$) were pooled every five samples for protein detection. The extracted protein was subjected to BCA protein concentration assay by BCA kit (Thermo Scientific, Waltham, MA, USA). Standard curves were plotted based on the optical density values of standards, which allowed the determination of protein concentration. SDS-PAGE was used for protein separation, followed by staining with Coomassie Brilliant Blue, then a fully automated digital gel image analysis system was applied for scanning.

The remaining samples were protease-digested, desalted, and then analyzed by mass spectrometry under the following conditions: capillary voltage of 1.5 KV, desiccant temperature of 180 °C, drying gas flow rate of 3.0 L/min, mass spectrometry scan range of 100–1700 m/z, ion mobility range of $0.75-1.4$ Vs/cm$^2$, collision energy range of 20–59 eV, followed by evaluation of raw data, database search, and screening for differentially expressed proteins. Gene Ontology (GO) and Kyoto Encyclopedia of Genes and Genomes (KEGG) databases were used to explore the associated functions and/or pathways of target proteins.

## Statistical analysis

Data analysis was performed using SPSS 29.0 (IBM Corp., Armonk, NY, USA, Version 29.0). Quantitative data were expressed as means $\pm$ SD, while categorical variables were presented as percentages. Two independent samples were compared using Student's $t$-test or the Mann–Whitney U-test (when variances were homogeneous and non-normal), and chi-square tests or Fisher's exact probability tests were applied for count data. $P < 0.05$ was considered statistically significant.

Proteins with a fold change >1.2 or <0.83 and $P < 0.05$ were considered differentially expressed proteins (DEPs). The DEPs were further subjected to GO and KEGG enrichment analysis using the R package. The results of the enrichment were visualized the ggplot2 package in R. Protein interaction analysis was performed using String.

# RESULTS

## Characteristics of patients

A total of 226 patients were enrolled. Two patients were excluded due to perioperative deaths. By the end of 6 weeks, 22 patients had recovered their cognitive ability compared with the second day after surgery (out of 53 patients with PND on the second day post-surgery, 44 on the day before discharge, and 31 by the sixth week). Eighteen cases were lost to follow-up. A total of 184 patients were included finally. Among these patients, 31 patients

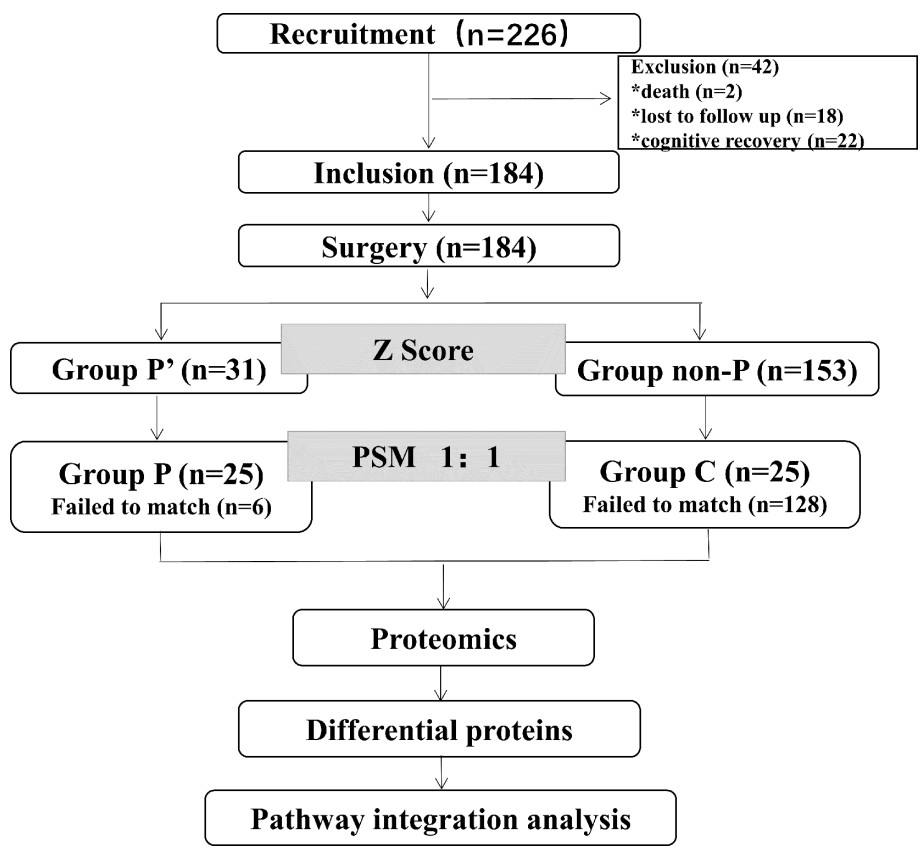

**Figure 1 Flow diagram showing the recruiting criterion.**

(group P') developed PND (16.8%), while 184 patients (group non-P) did not develop PND. Twenty-five patients were successfully matched in the group P and 25 patients were matched in the group C through PSM (Fig. 1). Detailed demographic characteristics of the healthy group control (spouse of patients, $n = 50$) and the surgical group ($n = 226$) before surgery are presented in Table S1.

## Risk factors of PND

Through a univariate analysis of preoperative, intraoperative, and postoperative related indicators for both the non-P and P' groups, it was found that age, the presence of chronic disease, the dose of sufentanil, and the duration of CPB are factors influencing the occurrence of PND refer to Table 1. The MMSE scores and MoCA scale scores of the non-P and P' groups on the day before surgery, the second day after surgery, the day before discharge, and 6 weeks after surgery are presented in Table S2. The comparisons of relevant parameters between the preoperative, intraoperative, postoperative group non-P and group P' are detailed in Tables S3, S4, and S5. The MMSE and MoCA scale scores of the group P and C after matching can be found in Table S6. The comparisons of relevant parameters before surgery, during surgery and after surgery between the group P and C after PSM can be seen in Table S7.
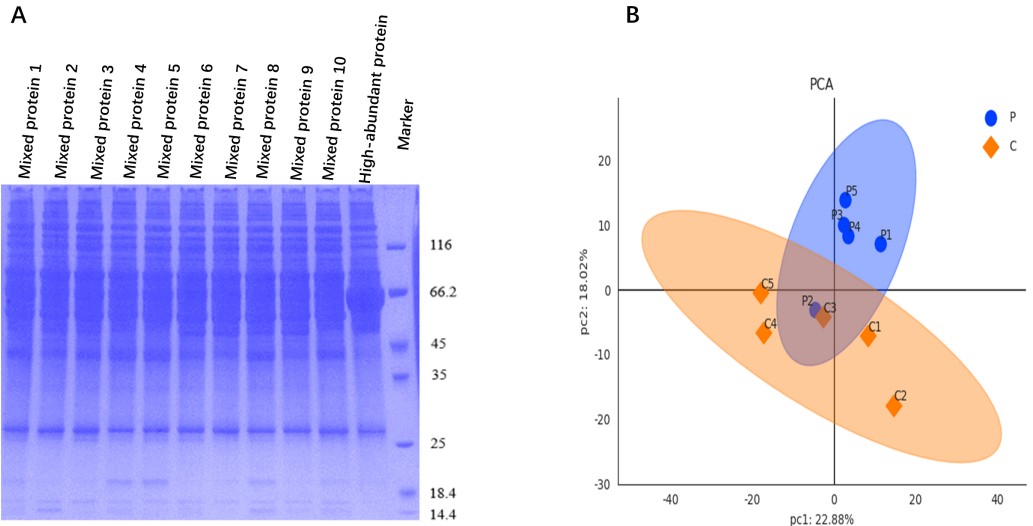

**Figure 2** **Protein quality control and data quality control.** (A) SDS-PAGE image shows clearly resolved and distinguished protein bands for samples from groups P (mixed proteins 1–5) and C (mixed proteins 6–10), with a clean background. (B) PCA was used to visualize the clustering trend between groups P and C, revealing distinct protein profile separation.

## Protein quality control and data quality control

Upon examination *via* sodium dodecyl sulfate-polyacrylamide gel electrophoresis (SDS-PAGE), the protein bands of each sample were clearly resolved and distinguishable. The distribution characteristics of these protein bands exhibited similarity among the various samples, and the gel background remained clean and clear, as shown in Fig. 2A.

Raw data were retrieved from the database, and median normalization and $\log_2$ logarithm transformation were applied to obtain reliable proteins. Utilizing the expression level of reliable proteins for principal component analysis (PCA), we obtained a good similarity and difference between group P and C, as shown in Fig. 2B.

## Screening DEPs

After analyzing the serum of the successfully matched group P and C on the second day after surgery using 4D-Label free serum proteomics, 37 differential proteins were identified, with six proteins were up-regulated and 31 proteins were down-regulated. Subsequent GO and KEGG analyses are illustrated in Fig. 3.

## Analysis of targeted proteins

The 37 differential proteins were cross-referenced with "PND", "POCD", and "cognitive" to select proteins related to cognitive function. These were listed in ascending order of *P*-value, resulting in 10 up-regulated and 2 down-regulated proteins. See Table 2 for details. The relationship between these DEPs and MMSE scores from this study was analyzed and ordered according to *P*-value, as shown in Table 3.

**Table 1** Comparison of parameters between group non-P and group P' before, during, and after surgery.

| Parameter | group non-P (*n* = 153) | group P' (*n* = 31) | *P*-value |
|---|---|---|---|
| **Preoperative indicators** | | | |
| Gender (%) | | | |
| Female/Male | 72/81 | 11/20 | 0.076 |
| Age (y) (%) | | | |
| 45∼54/55∼64/65∼75 | 78/49/26 | 10/10/11 | 0.043[*] |
| BMI (mean ±SD) | 22.02 ± 3.46 | 22.00 ± 3.10 | 0.974 |
| Education (%) | | | |
| 3/6/9 years | 44/45/55 | 9/6/12 | 0.439 |
| MMSE score pre-operation | 27.43 ± 1.64 | 27.22 ± 1.98 | 0.540 |
| Chronic diseases (at least one of COPD, vascular disease, anemia or arthritis) | | | |
| No/Yes | 125/28 | 19/12 | 0.014[*] |
| **Hypertension** | | | |
| No/Yes | 163/17 | 26/5 | 0.630 |
| Stage 1/Stage 2/Stage 3 | 10/4/3 | 2/2/1 | 0.729 |
| Age of hypertensive patients | 58.94 ± 7.85 | 63.00 ± 10.49 | 0.306 |
| MMSE score pre-operation | 26.41 ± 1.70 | 26.8 ± 2.17 | 0.676 |
| MoCA score pre-operation | 25.00[24.00, 26.00] | 24.00[23.50,26.50] | 0.542 |
| Years with hypertension | 4.00[2.00, 6.50] | 3.00[1.50, 7.00] | 0.968 |
| **Intraoperative indicators** | | | |
| Sufentanil (ug)(mean ±SD) | 327.56 ± 55.68 | 348.55 ± 39.52 | 0.047[*] |
| Propofol (ml) (mean ±SD) | 148.26 ± 35.45 | 153.06 ± 38.55 | 0.499 |
| Midazolam(mg)(mean ±SD) | 4.62 ± 1.90 | 5.13 ± 1.38 | 0.160 |
| Etomidate (mg)(mean ±SD) | 15.20 ± 3.36 | 16.19 ± 3.28 | 0.132 |
| Dexmedetomidine (%) | | | |
| No/Yes | 98/54 | 15/16 | 0.093 |
| CPB time (min)(mean ±SD) | 147.09 ± 46.38 | 165.35 ± 48.11 | 0.048[*] |
| HR (bmp) (mean ±SD) | 92.27 ± 8.60 | 92.33 ± 8.82 | 0.973 |
| MAP (mmHg)(mean ±SD) | 73.76 ± 3.41 | 73.77 ± 4.01 | 0.987 |
| BIS(mean ±SD) | 46.84 ± 3.04 | 46.93 ± 3.21 | 0.884 |
| **Postoperative indicators** | | | |
| Intubation time in ICU (h) (IQR) | 19.00[14.00, 25.50] | 20.00[16.00, 29.00] | 0.140 |
| Dopamine (mg) (IQR) | 540.00[300.00,900.00] | 600.00[390.00,720.00] | 0.876 |
| Epinephrine (mg)(IQR) | 6.00[2.50, 7.75] | 5.00[2.85, 6.95] | 0.676 |
| VAS score (IQR) | 4.00[4.00, 5.00] | 4.00[4.00, 5.00] | 0.426 |

**Notes.**

[*]In comparison to the group non-P, $P < 0.05$.

IQR, Interquartile Range; BMI, Body Mass Index; COPD, Chronic Obstructive Pulmonary Disease; VAS, Visual Analogue Scale.

## GO and KEGG analyses

GO and KEGG analyses were performed on the differential proteins, followed by PPI analysis. GO enrichment analysis demonstrated significant alterations in cellular

Peer J

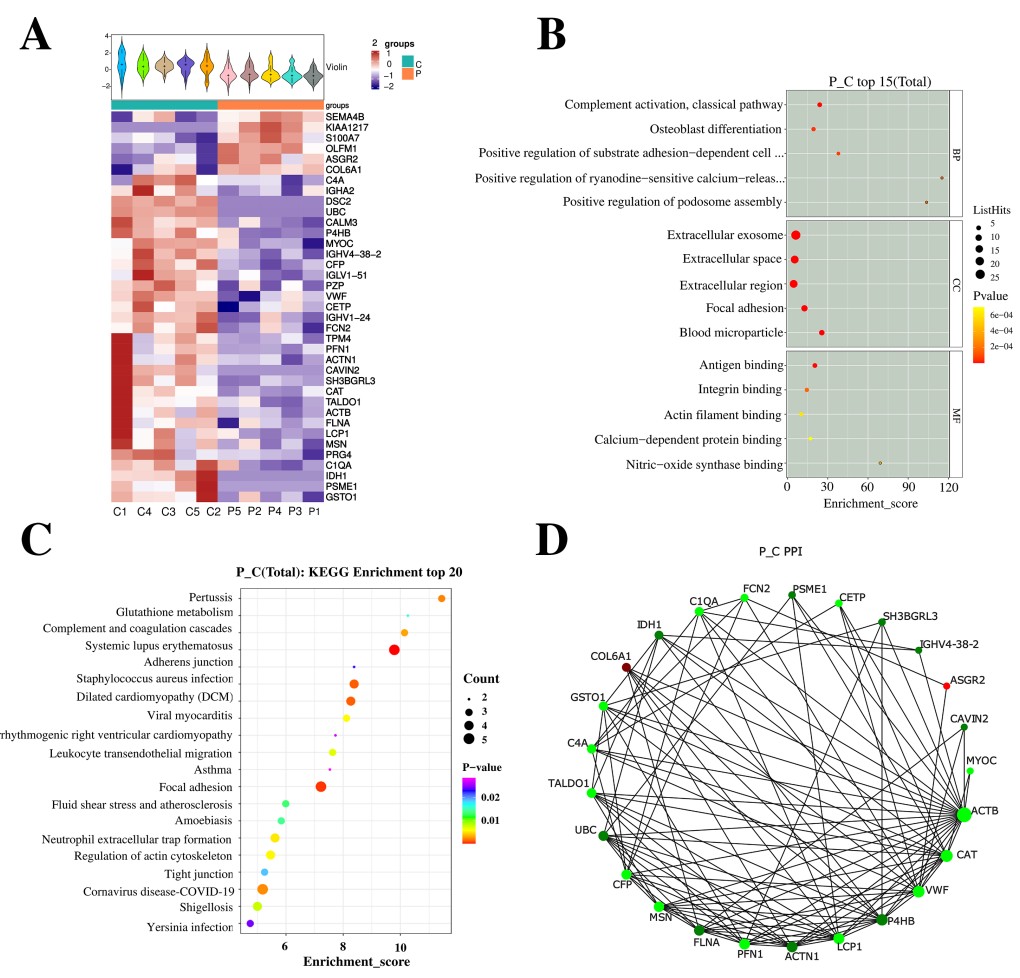

**Figure 3  Profiling and integrated analysis of DEPs between groups P and C revealed proteins linked to PND occurrence after cardiac valve surgery.** (A) The heatmap shows hierarchical clustering of quantitative protein data from 10 samples. (B) GO analysis, the figure displays Enrichment Score on the *x*-axis and top 5 BP/CC/MF terms on the *y*-axis. Larger bubbles denote more proteins and the color shift from yellow to red indicates increasing significance with smaller *p*-values. (C) Bubble diagram shows Enrichment Score *vs* top 20 pathways. Larger bubbles and red-to-purple gradient signify more differential proteins and increasing significance. (D) NetworkX visualized the top 25 connected nodes, with circles representing differential proteins color-coded by upregulation (red) or downregulation (green).

components of the cognitive-related DEPs. Majority of these DEPs were located in the extracellular exosomes and extracellular regions, with some also present in cytoplasmic vesicles and blood microparticles. KEGG enrichment analysis identified a total of 12 differential proteins with Listhits ≥2, sorted in ascending order of *p*-values, which were found to participate in glutathione metabolism, pertussis, peroxisome, and complement activation pathways. Refer to Fig. 4 for details.

**Table 2 DEPs associated with cognitive function after database search.**

|  | Accession | Gene name | FC | *P*-value |
|---|---|---|---|---|
| Down | O75874 | IDH1 | −0.0212 | 0.0000 |
| Down | P0DP25 | CALM3 | −0.8384 | 0.0025 |
| Down | P04275 | VWF | −0.7607 | 0.0072 |
| Down | P20742 | PZP | −0.7277 | 0.0140 |
| Down | P02745 | C1QA | −0.7079 | 0.0204 |
| Down | P04040 | CAT | −0.6994 | 0.0255 |
| Down | P07737 | PFN1 | −0.6848 | 0.0277 |
| Down | P26038 | MSN | −0.6558 | 0.0308 |
| Down | P78417 | GSTO1 | −0.6538 | 0.0318 |
| Down | P11597 | CETP | −0.6478 | 0.0335 |
| Up | Q9NPR2 | SEMA4B | 0.5946 | 0.0449 |
| Up | P12109 | COL6A1 | 0.6443 | 0.0498 |

**Table 3 Correlation between DEPs and MMSE scores.**

|  | Accession | Gene name | *P*-value | MMSE corr | MMSE corr_*P* value |
|---|---|---|---|---|---|
| Down | P0DP25 | CALM3 | 0.0025 | −0.8384 | 0.0024 |
| Down | O75874 | IDH1 | 0.0000 | 0.8200 | 0.0037 |
| Down | P04275 | VWF | 0.0072 | −0.7607 | 0.0106 |
| Down | P02745 | C1QA | 0.0204 | −0.7277 | 0.0170 |
| Down | P78417 | GSTO1 | 0.0318 | −0.7079 | 0.0220 |
| Down | P20742 | PZP | 0.0140 | −0.6994 | 0.0244 |
| Down | P26038 | MSN | 0.0308 | −0.6848 | 0.0289 |
| Down | P07737 | PFN1 | 0.0277 | −0.6558 | 0.0395 |
| Down | P11597 | CETP | 0.0335 | −0.6538 | 0.0403 |
| Down | P04040 | CAT | 0.0255 | −0.6478 | 0.0428 |
| Up | P12109 | COL6A1 | 0.0498 | 0.6443 | 0.0444 |
| Up | Q9NPR2 | SEMA4B | 0.0449 | 0.5946 | 0.0698 |

## DISCUSSION

The incidence of PND after cardiac valve replacement surgery in this study is 16.8%. In addition to the risk factors consistent with previous studies, including age, sufentanil dosage, and duration of CPB. We also found that the presence of chronic diseases is a risk factor for the development of PND. Through DEPs screening and correlation analysis with cognitive scores, we identified 10 down-regulated differential proteins and two up-regulated differential proteins (Table 3). GO enrichment analysis revealed significant changes in cellular components after cardiac surgery, while KEGG enrichment analysis identified two pathways related to oxidative stress: the Glutathione S-transferase Omega-1 (GSTO1) and IDH1-participated glutathione metabolism pathway, and the IDH1 and CAT-enriched peroxisome pathway.

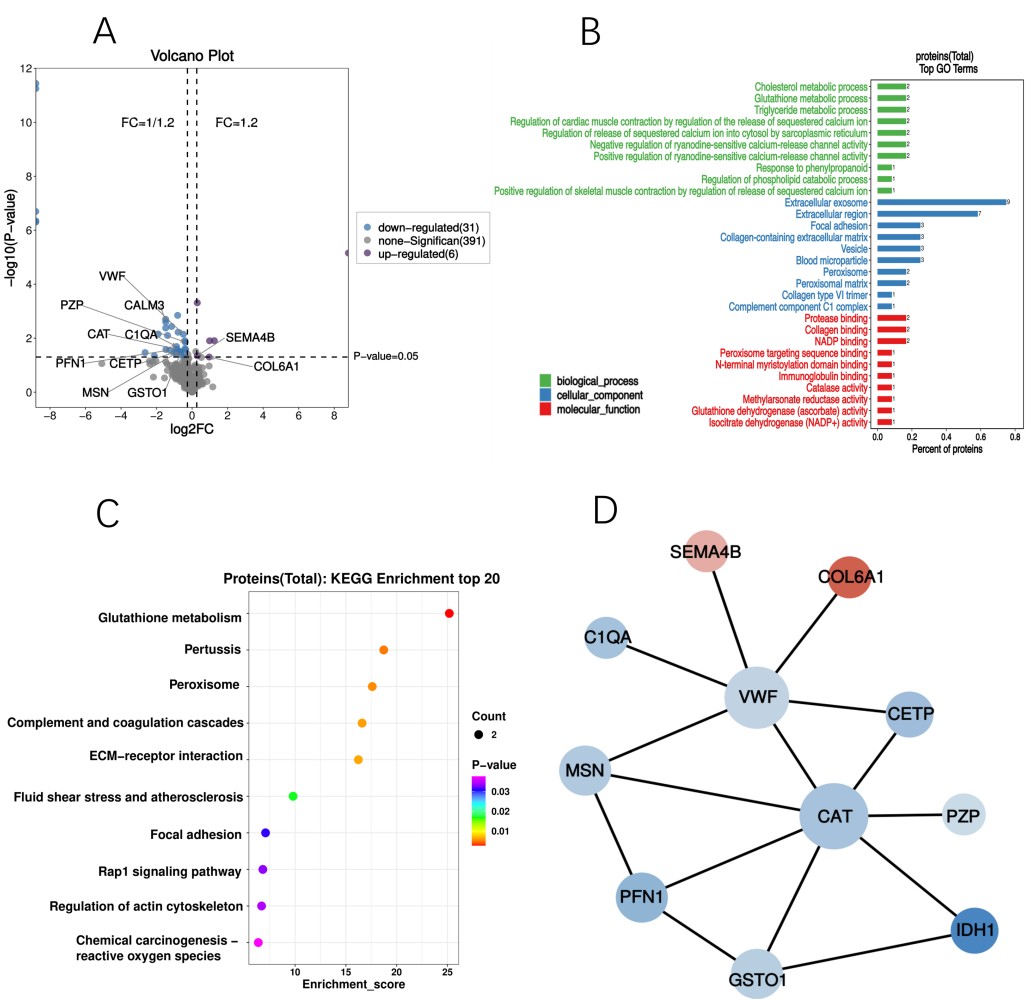

**Figure 4** **Profiling of cognition related DEPs.** (A) Volcano plot of DEPs; The blue dots in the figure represent down-regulated DEPs, while the purple dots represent up-regulated DEPs. (B) GO analysis of DEPs; Significant changes have occurred in the cellular components, with a majority of the DEPs primarily located in the extracellular exosomes and extracellular regions. (C) Bubble diagram of KEGG; Based on significance, DPEs are enriched in glutathione metabolism, pertussis, peroxidase, and complement activation pathways. (D) PPI network of DEPs; circles denote differential proteins (upregulated in red, downregulated in green), with size indicating connectivity level (larger circles = higher connectivity).

## Incidence of PND

Previous studies have shown that about 50∼70% (*Newman et al., 2006*) of patients exhibit a decline in cognitive ability one week after coronary artery bypass grafting (CABG), with 13∼40% (*Gao et al., 2005*; *Evered et al., 2009*) of patients continuing for one year. Moreover, the incidence rate of neurocognitive disorders following cardiac valve replacement is reported to be around 22%∼71% (*Ballester et al., 2011*; *Xia, Huang & Ansley, 2006*). These data consistently indicate a high incidence of PND following cardiac surgery. However, in the current study, the incidence rate of postoperative disorders was relatively lower (16.8%). There may be two reasons: 1. Our assessment of PND adopted

a Z-score less than −1.96 (equivalent to a decrease of 2SD), while some literatures used a Z-score less than 1SD (*Ackenbom et al., 2021*; *Deiner et al., 2021*; *Dustin Boone et al., 2022*; *Glumac, Kardum & Karanovic, 2018*), which people with less severe cognitive decline were also diagnosed as PND patients, which increased the incidence; 2. During the case collection, we excluded elderly patients with a surgical history, as well as those suffering from metabolic diseases (such as diabetes) or endocrine disorders (such as lipid disturbances), as these conditions could potentially affect cognitive ability (*Moller et al., 1998*; *Fabbri et al., 2015*; *Kim, Park & An, 2019*). This led to the study subjects being younger and having fewer underlying diseases, resulting in a lower incidence rate of PND compared to the above data. Although the proportion of elderly individuals was low in this study, age remained higher in the PND group ($P < 0.05$) than in the control group (C group) when analyzing the factors influencing PND occurrence.

It is well-known that aging is a natural, irreversible life process, inevitably accompanied by the decline of bodily systems, such as brain atrophy and the reduction of brain cells. Furthermore, with advancing age, chronic low-grade neuroinflammation is a hallmark of cognitive decline associated with the normal aging process (*Luo et al., 2019*). In these circumstances, cardiac valve patients in this study following anesthesia and surgery showed that older patients have a higher probability of developing PND, which is a risk factor for PND. This is consistent with the prevailing view that older age is an independent risk factor for PND (*Kubota et al., 2018*).

## Risk factors for PND

In our study design, we did not exclude patients with valvular heart disease who also have hypertension. Epidemiological studies have shown a significant correlation between hypertension and a decline in cognitive abilities, mild cognitive impairment, and dementia (*Kivipelto et al., 2001*; *Freitag et al., 2006*; *Gottesman et al., 2017*; *Mahinrad, Sorond & Gorelick, 2021*). Additionally, Iadecola's review highlighted that hypertension could affect various cognitive domains, including attention and executive functions. These impacts are often reflected in scores from comprehensive cognitive assessment tools such as MMSE and MoCA, typically resulting in lower scores (*Iadecola & Gottesman, 2019*). Upon reviewing relevant literature, we noticed that most studies exploring the relationship between hypertension and cognition included vascular diseases or metabolic disorders related to dyslipidemia, with the studies often spanning a long duration (about 20 years), showing a significant correlation between hypertension and cognitive decline (*Kivipelto et al., 2001*; *Freitag et al., 2006*; *Gottesman et al., 2017*).

However, in this study, through comparative analysis between the P' group and the non-P group, we found that hypertension is not a risk factor for PND. Additionally, our further analysis on aspects such as preoperative MMSE and MoCA scores, hypertension severity grading, age, and years with hypertension among the two groups of hypertensive patients showed no statistical difference (see Table 1). Thus, it can be inferred that the baseline characteristics related to hypertension were consistent between the two groups in this study. Beyond hypertension, we excluded diseases that could affect cognitive functions, such as diabetes, lipid disorders, central nervous system diseases and cerebrovascular diseases.

Nonetheless, we discovered that some PND patients still had comorbidities, including respiratory system diseases (*e.g.*, chronic obstructive pulmonary disease), digestive system diseases (*e.g.*, gastric diseases), cardiovascular system diseases (*e.g.*, coronary artery stenosis), and immune system diseases (*e.g.*, arthritis). Therefore, we defined chronic diseases as including at least one of the aforementioned diseases (*Monastero et al., 2007*) for our study. The reasons for decreased cognitive ability in patients with one or multiple chronic diseases are still not entirely clear. We speculate that the severity of the disease, medication, and interactions among diseases might mediate the occurrence of PND. This data is influenced by many confounding factors, which further clinical studies are required to clarify the specific causes (*Blaum, Ofstedal & Liang, 2002*). From our study, we noticed that this decreased cognitive ability coexists with patients' chronic diseases, which leads to a long-term inflammatory state in the body or cellular damage causing a decline in cognitive ability (*Tangestani Fard & Stough, 2019*). These patients often take medication to alleviate their symptoms. According to relevant studies, there is a steady increase in risk when using 1–4 kinds of drugs to five kinds of drugs, with the number of drugs taken showing a dose–response relationship with the risk of cognitive impairment (*Monastero et al., 2007*; *Shinohara & Yamada, 2016*). Patients who take more than four kinds of drugs have three times the risk of cognitive impairment compared to those who take no medication (*Blaum, Ofstedal & Liang, 2002*) . However, in this study, the coexistence of these chronic diseases did not result in an MMSE score of less than 24 on the day before surgery, thus they were not excluded. But through the analysis of related variables in the cognitive impairment and non-cognitive impairment groups after cardiac valve replacement surgery, chronic diseases still could be potential risk factors influencing the occurrence of PND.

## GSTO1

In the most enriched KEGG pathways, the glutathione metabolism pathway is significantly enriched ($P = 0.0027$) (Fig. 4C), which includes GSTO1 and IDH1 proteins. GSTO1is biologically related to late-onset Alzheimer's disease (AD), serving as a risk factor for cognitive impairment (*Wongtrakul et al., 2018*). GSTO1 is an omega class subtype of glutathione S-transferases (GST), and genetic variants of the omega class GST genes are related to the age of onset of diseases such as AD, Parkinson's disease, amyotrophic lateral sclerosis, and vascular dementia. GSTO1 is a protein with various cellular functions, participating in primary metabolism, detoxification and protection, resistance to oxidative damage, and sequestration of xenobiotics. GSTO1 at the mRNA level is related to several central nervous system traits, such as caudate glial fibrillary acidic protein levels, cortical gray matter volume, and hippocampal mossy fiber pathway volume. It is also found to be related to AD susceptibility genes, such as APP, Grin2b, Ide, Psenen, *etc* (*Wongtrakul et al., 2018*). It has been reported that GSTO1 expression levels are significantly reduced in patients with cognitive impairment (*Li et al., 2003*), and a significant correlation between GSTO1 polymorphism and the age of AD onset has been found (*Zhang et al., 2018*). It has been confirmed that GSTO1 gene knockout significantly impacts the expression of its downstream gene Pa2g4, and these two genes interact with other genes in the network during the development of AD (*Jia et al., 2022*). In our study, we found that in addition to

GSTO1′s antioxidative stress function in the redox system, IDH1, CAT and PFN1 are also related to oxidative stress.

## IDH1

IDH1, an isozyme of the IDH gene family, catalyzes the production of Nicotinamide Adenine Dinucleotide Phosphate (NADPH), which acts as a donor of reductive hydrogen within the body, participating in cellular resistance to oxidative stress responses. It has been proven that IDH1 is upregulated in an oxidative stress environment to limit oxidative damage (*Wahl et al., 2017*). The IDH1 mutation related to oxidative stress is a metabolic enzyme in the glycolytic pathway that catalyzes the oxidative decarboxylation of isocitrate to 2-oxoglutarate, providing cellular protection against oxidative stress (*Hodges et al., 2013*). Recently, *Walker et al. (2022)* utilized the GeoMx™ Digital Spatial Profiler (DSP) technology to study protein expression differences between individuals with dementia and AD neuropathology *versus* recovery individuals, and found that IDH1 expression was lower in the environment of resilient neurons with neurofibrillary tangles (nft). Similarly, in the present study, the expression of IDH1 was down-regulated which is consistent with *Walker et al.*'s *(2022)* research results.

Moreover, we observed that both IDH1 and CAT proteins were enriched in the transport and catabolic metabolism (Classification_level2) KEGG pathway, specifically the peroxisome pathway. The peroxisome pathway plays a pivotal role in cellular metabolic conversions, participating in various biological processes. It generates substantial energy and oxidative products through oxidative reactions and is involved in several cellular metabolic processes facilitated by oxidative reactions catalyzed within peroxisomes. The functionality of the peroxisome pathway largely depends on the intricate interplay and modulation between a plethora of enzymes and metabolic substances within the peroxisomes, thus fulfilling the biological functions of the peroxisome pathway.

## CAT

Catalase (CAT) is an essential antioxidant enzyme that reduces oxidative stress by decomposing hydrogen peroxide in cells to produce water and oxygen. The deficiency or dysfunction of catalase is thought to be associated with the pathogenesis of many age-related degenerative diseases, such as diabetes, hypertension, AD, Parkinson's disease, bipolar disorder, schizophrenia, and cancer (*Al-Abrash, Al-Quobaili & Al-Akhras, 2000*; *Jimenez-Fernandez et al., 2022*; *Lane, Wang & Lin, 2023*). Animal studies have also confirmed that 24 h after fracture surgery, the activity of CAT in the prefrontal cortex and hippocampus significantly decreases and remains low on the seventh day postoperatively in the hippocampus (*Netto et al., 2018*).

## PFN1

Interestingly, by examining the aforementioned differential proteins in the database, we also identified a connection between the PFN1 protein and oxidative stress. PFN1 (profilin 1) is a small protein composed of 140 residues that functions to regulate actin polymerization in cells. While PFN1 and SOD1 do not share functional similarities, their mutated forms display comparable patterns in neuropathology, particularly concerning

the ongoing progression of motor neuron degeneration (*Lim, Kang & Song, 2017*). Mutant variants of SOD1 have been linked to cellular membranes, especially the mitochondria and endoplasmic reticulum (*Israelson et al., 2010*). This aberrant integration into the ER membrane induces ER stress observed in ALS (*Sun et al., 2015*). In a similar way, *Lim, Kang & Song (2017)* speculated that certain mutations in PFN1 might trigger stress reactions associated with oxidative stress due to abnormal interactions with membranes, possibly representing one of the pivotal mechanisms in ALS pathogenesis.

Undoubtedly, the onset of PND is not solely attributed by oxidative stress. Based on the KEGG enrichment analysis, it is associated with the complement coagulation system, the Pertussis pathway, and the Rap1 signaling pathway.

## LIMITATIONS

In our endeavor to identify differential proteins associated with PND occurrence following cardiac valve replacement, we recognize several areas that merit further exploration. Firstly, while our rigorous inclusion criteria, exclusion criteria and statistical methods helped minimize preoperative confounding factors, the inclusion of a preoperative serum proteomic profile as a baseline comparison would have enhanced the reliability of the identified proteins. Secondly, the current study focused primarily on the discovery phase, identifying potential protein markers. However, we acknowledge that functional validation through animal or cellular models is a crucial next step in confirming the biological significance of these proteins. Future research efforts should aim to bridge this gap, providing a more comprehensive understanding of the roles these proteins play in the pathophysiology of PND.

## CONCLUSION

The prevalence of PND patients six weeks after heart valve replacement surgery stands at 16.8%. The occurrence of PND may be mainly related to the expression levels of oxidative stress-related proteins: GSTO1, IDH1, CAT, and PFN1. The occurrence of PND can impact principal metabolic pathways, such as glutathione metabolism and peroxisome pathway, in clinical patients.

### Funding

This study was funded and supported by the Zunyi Science and Technology Plan Project (Grant No: Zunyi City Kehe HZ (2022) 331) and the Guizhou Provincial Health Commission (Grant No: gzwkj2022-129). The funders had no role in study design, data collection and analysis, decision to publish, or preparation of the manuscript.

### Grant Disclosures

The following grant information was disclosed by the authors:
The Zunyi Science and Technology Plan Project: Zunyi City Kehe HZ (2022) 331.
The Guizhou Provincial Health Commission: gzwkj2022-129.

## Competing Interests

The authors declare there are no competing interests.

## Author Contributions

- Huanhuan Ma conceived and designed the experiments, performed the experiments, analyzed the data, prepared figures and/or tables, and approved the final draft.
- Yiyong Wei conceived and designed the experiments, analyzed the data, authored or reviewed drafts of the article, and approved the final draft.
- Wei Chen performed the experiments, analyzed the data, prepared figures and/or tables, and approved the final draft.
- Song Chen performed the experiments, prepared figures and/or tables, and approved the final draft.
- Yan Wang analyzed the data, prepared figures and/or tables, and approved the final draft.
- Song Cao analyzed the data, authored or reviewed drafts of the article, and approved the final draft.
- Haiying Wang conceived and designed the experiments, authored or reviewed drafts of the article, and approved the final draft.

## Human Ethics

The following information was supplied relating to ethical approvals (*i.e.,* approving body and any reference numbers):

The Biomedical Research Ethics Committee of the Affiliated Hospital of Zunyi Medical University approved the study (2022 (KLL-2022-700)).

## Clinical Trial Ethics

The following information was supplied relating to ethical approvals (*i.e.,* approving body and any reference numbers):

Chinese Clinical Trials Registry 2022 (ChiCTR2200064929)

## Data Availability

https://proteomecentral.proteomexchange.org/cgi/GetDataset?ID=PXD048818

## Clinical Trial Registration

The following information was supplied regarding Clinical Trial registration:

Chinese Clinical Trials Registry 2022 (ChiCTR2200064929)

## Supplemental Information

Supplemental information for this article can be found online at http://dx.doi.org/10.7717/peerj.17536#supplemental-information.

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
