# Peer review of "Serum proteomics study on cognitive impairment after cardiac valve replacement surgery: a prospective observational study"

_PeerJ, doi:10.7717/peerj.17536_

## Round 0.1 · original submission · Major Revisions

Dear authors,

The study entitled “Serum proteomics study on cognitive impairment after cardiac valve replacement surgery: a prospective observational study” demonstrated excellent findings using an appropriate methodological approach. However, some important points must be clarified in the manuscript. Your article has great potential for publication on PeerJ, but the reviewers have requested substantial changes to be made, mainly in methodology and discussion sessions.

·

Basic reporting

The study entitled “Serum proteomics study on cognitive impairment after cardiac valve replacement surgery: a prospective observational study” is a prospective observational study to observe serum proteomics differences in perioperative neurocognitive disorders patients (PND) after cardiac valve replacement surgery. The main finds of this study were a) incidence of PND was 16.8%; b) Age, chronic illness, sufentanil dosage, and time of cardiopulmonary bypass are risk factors for PND; identification of 31 down-regulated proteins and 6 up-regulated proteins and d) changes in cellular components after cardiac surgery related to two oxidative stress pathways: the 227 Glutathione S-transferase Omega-1 (GSTO1) and IDH1-participated glutathione metabolism 228 pathway, and the IDH1 and CAT-enriched peroxisome pathway. The authors list as a limitation the fact that they did not exclude hypertensive patients from the study.
Introduction is concise and relevant. The objectives are clear.
However, the raw data provided should be better identified and written entirely in English.

Experimental design

The study is related to Medical or Health sciences within the scope of the PeerJ.
The study aims to characterize and extend knowledge about post-surgical cognitive impairment, specifically after heart valve replacement surgery.
In the item 2.2 (Anesthesia and surgery) and 2.3 (Neuropsychological test), the verb tenses must be corrected.
Additionally, the authors should also provide the English language version of the Ethical material and Informed Consent Form (Supplemental data).
The legends in figure 4 are mostly illegible and must be rewritten.
In Table 1, “Preoperative indictors” must be corrected and bolded.

Validity of the findings

In general, the work methodology is appropriate, and the results are clear. However, for us to better understand the implications of the findings, two main questions need to be answered:
- Considering that Moller at al. (DOI: 10.1016/s0140-6736(97)07382-0) have reported that after surgery cognitive disfunctions may be present, the authors of the present work have considered the history of surgeries of the patients and a possible additive effect in neurocognition?
- Have the authors considered that protein expression could be altered before the cardiac valve replacement surgery? Is there any control of it to be presented?

Additional comments

No comments.

·

Basic reporting

The study entitled “Serum proteomics study on cognitive impairment after cardiac valve replacement surgery: a prospective observational study” aimed to use proteomics to explore potential Perioperative Neurocognitive Disorder-related proteins by analyzing serum samples from patients undergoing post cardiopulmonary bypass valve replacement. The authors conducted a prospective cohort study with 184 patients who underwent heart valve replacement at Zunyi Medical University Hospital (China).
The main finds of this present study were that:
- Of the 184 patients, 31 (16.8%) had an incidence of Perioperative Neurocognitive Disorder after cardiac valve replacement surgery;
- Age, sufentanil dosage, duration time of post-cardiopulmonary bypass, and chronic diseases are risk factors for the development of Perioperative Neurocognitive Disorder;
- 10 down-regulated differential proteins and 2 up-regulated differential proteins were identified;
- Were showed significant changes in cellular components after cardiac surgery, two pathways related to oxidative stress were identified: the Glutathione S-transferase Omega-1 and IDH1-participated glutathione metabolism pathway, and the IDH1 and CAT-enriched peroxisome pathway.

Experimental design

The experimental design of this study is adequate. However, In Table 1 is missing the letter "a" in the term "Preoperative indicators". The Figure 4 should be improved, it is very difficult to read and interpret.

Validity of the findings

- The sutdy methodology is appropriate. However, there is a doubt regarding the sample of patients in this study. The authors comment in the study limitations section that they were not excluded patients with hypertension due to the difficulty in obtaining an adequate number of clinical cases. In this case, did the authors check the antihypertensive medications of these patients? If yes, it would be wise to mention in the methods section. If the authors did not perform this procedure, could you say whether such medications could influence your results? Another point is, could hypertension itself trigger changes in these proteins? The authors could give more emphasis to these points in the discussion of the manuscript.
- The authors made a comparison of some parameters (Preoperative indicators and Intraoperative indicators) between the group non-P and group P (before, during, and after surgery). In this sense, have the authors analyzed serum proteomics differences in perioperative neurocognitive disorders patients before the cardiac valve replacement surgery? Can the authors say with certainty whether these proteins would not be altered before surgery just with the experimental groups you used? The authors could discuss this topic in more detail in the study discussion.

Additional comments

No comments

---

## Round 0.2 · Minor Revisions

Dear authors, you did a good job at answering the questions of reviewers and improving the manuscript. However, there are still some remaining concerns the reviewers with the manuscript.

·

Basic reporting

As previously stated, the introduction is concise and relevant. The text is professionally written. In the revised version, the raw data has been revised and the tables are understandable.

Experimental design

The text was corrected. Reviewed raw data were provided.

Validity of the findings

In the revised text, the authors reported that surgical history was one of the exclusion criteria. This information was not included in the first version of the manuscript. Then, I ask the authors whether all inclusion and exclusion criteria are properly described in the revised version of the manuscript.

Additional comments

No additional comments.

·

Basic reporting

The authors did a good job on the new revised version of this manuscript. The text has been appropriately corrected and the reviewed data were provided.

Experimental design

In the revised version, the tables and figures are better now understandable.

Validity of the findings

While they authors cannot confidently state that the proteins studied would not be altered before surgery based solely on your experimental groups, it is now believed that your experimental design and control measures have minimized the potential impact of preoperative factors on your results. The authors have also discussed the potential limitations and implications of this aspect in the revised manuscript to provide a more comprehensive understanding of study findings.

Additional comments

No additional comments.

---

## Round 0.3 · accepted · Accept

Congratulations to the authors. I consider that all the considerations made in the review were addressed. And a significant improvement in the work can be observed.